# OpenReview forum: "Learning to Boost Resilience of Complex Networks via Neural Edge Rewiring"
_ICLR.cc/2023/Conference — Submitted to ICLR 2023_

### Official Review · Reviewer_sxfB · 2022-10-25

**Confidence:** 2
**Correctness:** 4
**Technical Novelty And Significance:** 3
**Empirical Novelty And Significance:** 3
**Recommendation:** 8

**Clarity, Quality, Novelty And Reproducibility:**

The paper is quite clear, however, adding a pseudocode in the appendix would be good. The idea in the paper is quite novel. The code is open sourced, so it is reproducible.

**Strength And Weaknesses:**

# Strengths:
1. The paper is well-written.
2. The experiment results show significant gain
***
# Weaknesses:
1. The algorithm can be hard to follow. It would be good to add a pseudocode in the Appendix.


**Summary Of The Paper:**

Current GNN models rely on strong features for good performance. However, they struggle in the absence of features and when only the topology is present. This is true particularly in the case of making network resilient to all forms of attack. The paper proposes to make network resilient even in the absence of availability of features. In addition, contrast to earlier approaches, this paper proposes a learning-based method. The key idea is to create a series of subgraph by removing the node with the highest degree and the learn to aggregate node embeddings from such subgraphs.

**Summary Of The Review:**

The paper proposes an approach to make network resilient even when the features are unavailable. The method seems to bear some resemblance to [A] but the goals were quite different. Regardless, the ideas are quite novel. The gains shown in the experiment section are pretty impressive. I am incline to accept the paper.
***
# References:
[A] GraphOpt: learning optimization models of graph formation, Rakshit Trivedi, Jiachen Yang, Hongyuan Zha, ICML 2020

---

### Official Review · Reviewer_aMne · 2022-10-25

**Confidence:** 3
**Correctness:** 3
**Technical Novelty And Significance:** 3
**Empirical Novelty And Significance:** 3
**Recommendation:** 3

**Clarity, Quality, Novelty And Reproducibility:**

Clarity: Unclear
- please refer to the weakness section for details.

Quality: Fair
- Though the experimental results are surprisingly good, there is not enough interpretation to explain why we can get this kind of result, which makes the quality fair.

Novelty: Good
- The studied topic of considering resilience tasks under an inductive setting is novel and interesting.

Reproducibility: Fine (between Fair to Good)
- The experimental setups for the proposed method are given in appendix C2, and experimental setups for baselines are in appendix C4.  So the hyper-parameter setting is fine for reproduction. But it would be better if the example code (not necessary to be the full code) on at least one dataset can be provided.

**Strength And Weaknesses:**

Strength:

1. The problem of considering resilience tasks with utility balancing is very interesting, and doing it under an inductive setting makes this work novel.
2. The experimental results are very impressive.

Weakness (and questions):
1. The writing is unclear and very hard to follow.
1) I think the authors abuse the appendix a little bit. I think usually people put the supplementary stuff (like experiments on extra datasets, related background information, details for the baseline models, etc) in the appendix, but would always keep the main paper self-contained. However, in this paper, the authors put some important details that would affect the reading in the appendix, i.e.the main paper itself is not self-contained and can not be fully understood without the appendix.
2) Sometimes the audience may need to check the many different sections to understand a single definition. For example, to understand what is a resilience metric in section 3, the audience needs to first go to section 5.1, then go to Appendix B.1, and then, the definition is still not very clear, because the authors use the term "certain attack strategy" without giving concrete examples or explaining the requirements of the attack strategy.
3) A small question for figure 1, why will AB and CD be selected at the t+2k+1 step?

2. The experimental results are not well-interpreted.
1) For table 1, in datasets BA-50 and BA-100, why do almost all the learning-based methods get 0(1)?
2) why is ResiNet capable of obtaining such a good performance boost on the EU dataset (>100% performance improvement on the second best)? what is so special about this dataset?
3) why are so many "x" in table 1?


**Summary Of The Paper:**

This work aims to study the problem of improving the resilience of a network while trading off between network utility. The authors point out that though edge rewiring can be a promising direction for network resilience improving, existing learning-free methods are not enough due to their limitations of transduction, local optimality, and utility loss issue; and existing learning-based GNN models are also not enough as they may fail the network resilience optimization. Therefore, the authors provide the solution ResiNet (which includes the newly developed FireGNN model for feature-less graphs), and evaluate it on rich benchmark datasets.

**Summary Of The Review:**

Though the novelty of this work is fine to me and the experimental results are impressive, I think the writing is very unclear, and the experimental results are not well-explained, therefore, I think it is not ready to get published.

---

### Official Review · Reviewer_6mo7 · 2022-10-26

**Confidence:** 4
**Correctness:** 3
**Technical Novelty And Significance:** 3
**Empirical Novelty And Significance:** 3
**Recommendation:** 5

**Clarity, Quality, Novelty And Reproducibility:**

The problem studied is not appeared yet and it's novel. The writing is in general good but may missing some details as asked before.

**Strength And Weaknesses:**

**Strength**:
1. Improving network resilience is an important real-world problem. And the method is the first learning based method for finding policies to optimize resilience.
2. The writing is easy to follow, with many visualization to help understanding the pipeline.
3. Experimental setup is solid for the new problem, with many traditional baselines, which should be considered as a contribution.

**Weakness**:
1. The author claimed the designing of FireGNN is effective for modeling the edge selection. The node deleting idea is already explored in [Cotta et al. 21] and [Bevilacqua et al. 21] for improving GNN expressivity. Also deleting by node degree is inherently give a specific ordering of the graph, and it's really strange to see that all other GNNs failed to do edge selection but only the proposed method can do. Even more strange, the biggest improvement is combing from K from 0 to 1, which is hard to understand. Notice that K increasing from 0 to 1 just uses one additional graph for each step, which is similar to using 2-tuple representation. So my intuition is that you need 2-tuple based GNN for edge selection like 2-WL or 2-FWL, and I recommend the author to test with PPGN [Maron et al. 19].
2. Similar to above comments, the GNN baselines picked are kind of out-of-date, and I suggest the author to consider include many recent more expressive GNNs.
3. Perhaps the biggest question is the setup of transductive setting and inductive setting. From my understanding, what we really care about should be the inductive setting. However the author only report partial result with the designed method only, and there is no comparison with non-learning based baselines. Hence I suspect whether the proposed method can really work under inductive setting, or whether comparable with non-learning baselines.
4. For transductive setting, it seems a bit unfair to compare learning based method and traditional non-learning based method, as the the learning based method needs extremely higher time to train/learn. So the result in Table 1 seems questionable. To give a fair comparison, the author should also report the computational time for all methods. Also I would like to ask whether transductive setting is meaningful in real-world, and I wish the author can give some example about when transductive setting can be used.
5. Again, the current inductive setting only uses simulation dataset, I wish to see its performance on real-world problem, giving that the problem itself is valuable because of its real-world application.





**Reference**:
[Cotta et al. 21] Reconstruction for Powerful Graph Representations
[Bevilacqua et al. 21] Equivariant subgraph aggregation networks
[Maron et al. 19] Provably Powerful Graph Networks


**Summary Of The Paper:**

The paper studies the problem of learning to improve network resilience and unity with reinforcement learning on graphs, which is a important combinatorial optimization problem giving its application in power system and other robust networks. The author designs a reinforcement learning framework with modeling the decision of edge rewiring sequentially with MDP. Policy network is modeled with improved graph neural network as many GNNs failed to model edge selection. The author demonstrate the effectiveness of designed RL based policy over simulation datasets and real-world datasets, on transductive setting and inductive setting.

**Summary Of The Review:**

The author studies the resilience optimization problem with edge rewiring through reinforcement learning and graph neural network. The problem itself is interesting and the pipeline designed is reasonable. My concerns are mainly about experimental settings and evaluation baselines.

---

### Official Review · Reviewer_CVfp · 2022-11-02

**Confidence:** 5
**Correctness:** 3
**Technical Novelty And Significance:** 3
**Empirical Novelty And Significance:** 3
**Recommendation:** 3

**Clarity, Quality, Novelty And Reproducibility:**

The clarity and quality of this paper is relatively good.

The authors propose the first (claimed by themselves) inductive learning-based approach, ResiNet, to boost the resilience of complex networks, and the design of FireGNN learns meaningful representations via the proposed filtration process. The novelty of this paper is also guaranteed.

The authors claim that their implementation has already been open sourced, but I find no external links for their codes. The reproducibility of the paper needs to be further confirmed.


**Strength And Weaknesses:**

Strength:
1. The authors propose the first (claimed by themselves) inductive learning-based approach, ResiNet, to boost the resilience of complex networks, and the design of FireGNN learns meaningful representations via the proposed filtration process.
2. The experimental results are detailed and plentiful, which provide many perspectives to understand and prove the effectiveness of model.
3. The authors provide in-depth analysis for their approach and the resilience task itself, further illustrating the reason and the validity of their proposed designs.

Weakness:
1. There are small writing mistakes in the manuscript, for example, the footnote in the 1st page containing repeating “network resilience”, which makes me confused. In Section 4.2 (5th page) there are wrong spelling “FirGNN”.

2. In Table 1, the authors show that many learning-free methods can also achieve competitive performance, however their computational cost are relatively low. The authors also need to show the computational cost, such as the number of parameters or the training time for their reinforcement learning approach to	demonstrate if it is worth trading the cost using the performance.

3. The generalization of reinforcement learning method is important. However, “the generalization on optimizing different utility and resilience metrics” does not seem to be surprising for a learning-based model. What I am concerned about is the performance of the model which trains on synthetic networks but tests on real-world networks, further indicating its generalization and real-world applications. After all, different sizes of	BA networks	also	share similar characteristics and experiments on them are not convincing.


**Summary Of The Paper:**

The resilience of complex networks is a critical structural characteristic in network science, measures the network’s ability to withstand noise corruption and structural changes. The authors propose a framework, ResiNet, combining a variant of GNN called FireGNN to model the structural features of networks and reinforcement learning, to enhance the resilience of network with moderate loss of network utility via neural edge rewiring, which selects two directional edges step by step and rearranges them among four involved nodes, connecting two original head nodes and two original tail nodes. Extensive experiments demonstrate the validity of ResiNet in many kinds of networks and generalization on optimizing different utility and resilience metrics.

**Summary Of The Review:**

The authors propose a framework, ResiNet, combining a variant of GNN called FireGNN to model the structural features of networks and reinforcement learning, to enhance the resilience of network with moderate loss of network utility via neural edge rewiring. The experimental results and analysis are also comprehensive. However, due to the weaknesses and concerns above, I think further experiment and revision are needed, so I recommend to reject.

---

### Decision · Program_Chairs · 2023-01-20

**Decision:**

Reject

**Justification For Why Not Higher Score:**

no rebuttal was provided

**Justification For Why Not Lower Score:**

N/A

**Metareview: Summary, Strengths And Weaknesses:**

The paper develops a network resilience optimization method with a topological rewiring approach. The paper is interesting, though of somewhat limited novelty as related approaches have been proposed before. Reviewers raised several concerns, which remained unaddressed as the authors did not provide a rebuttal. We therefore recommend rejection.